# Land-Use Carbon Emissions in the Yellow River Basin from 2000 to 2020: Spatio-Temporal Patterns and Driving Mechanisms

**DOI:** 10.3390/ijerph192416507

**Published:** 2022-12-08

**Authors:** Mingjie Tian, Zhun Chen, Wei Wang, Taizheng Chen, Haiying Cui

**Affiliations:** 1The College of Geography and Environmental Science, Henan University, Kaifeng 475000, China; 2School of Philosophy and Public Management, Henan University, Kaifeng 475000, China; 3School of Culture Industry and Tourism Management, Henan University, Kaifeng 475000, China; 4School of Public Administration, Central China Normal University, Wuhan 430079, China

**Keywords:** Yellow River Basin, land-use, carbon emission, spatio-temporal pattern, driving mechanism

## Abstract

In the context of global climate governance, the study of land-use carbon emissions in the Yellow River Basin is crucial to China’s “dual-carbon” goal in addition to ecological conservation and the high-quality developments. This paper computed the land-use carbon emissions of 95 cities in the Yellow River Basin from 2000 to 2020 and examined its characteristics with respect to spatio-temporal evolution and driving mechanisms. The findings are as follows: (1) The overall net land-use carbon emissions in the Yellow River Basin rose sharply from 2000 to 2020. (2) From a spatial perspective, the Yellow River Basin’s land-use carbon emissions are high in the middle-east and low in the northwest, which is directly tied to the urban development model and function orientation. (3) A strong spatial link exists in the land-use carbon emissions in the Yellow River Basin. The degree of spatial agglomeration among the comparable cities first rose and then fell. “Low–Low” was largely constant and concentrated in the upper reaches, whereas “High–High” was concentrated in the middle and lower reaches with an east-ward migratory trend. (4) The rates of economic development and technological advancement have a major positive driving effect. Moreover, the other components’ driving effects fluctuate with time, and significant geographical variance exists. Thus, this study not only provides a rationale for reducing carbon emissions in the Yellow River Basin but also serves as a guide for other Chinese cities with comparable climates in improving their climate governance.

## 1. Introduction

Rising greenhouse gas emissions, mostly that of carbon dioxide, have led to a series of environmental changes that seriously harm human survival, such as global warming, rising sea levels, increasingly frequent extreme weather events, and the rapid spread of viruses [1]. The global community is highly concerned about climate change. The signing and prompt implementation of the Paris Agreement indicate that the world is collectively and unwaveringly committed to transitioning into a global low-carbon economy. Currently, the largest developing nation in the world is China. Its rapid economic growth following the 1978 economic reforms and its subsequent opening up has led to a dramatic increase in carbon emissions, and it is currently the country with the highest carbon emissions in the world [2]. As a result, in the face of enormous expectations from countries across the globe, China has taken on the significant responsibility of global environmental regulation. It announced its aim of “peak carbon by 2030, carbon neutrality by 2060” at the UN General Assembly’s 75th session [3]. However, China is currently dealing with issues such as an underdeveloped industrial structure, subpar technological tools, and challenging development and transition. Thus, the path to achieving the “dual carbon” aim is besieged by numerous challenges [4].

Land is an essential component of human existence and development. Land-use is the most direct expression of the connection between humans and nature, which is of tremendous significance to the global climate and environmental change [5]. Land-use change directly or indirectly affects the carbon cycle of the terrestrial ecosystem by changing the original land cover type and the social and economic activities it carries [6]. In addition, climate and environmental change can also introduce extra pressures with respect to the land [7] and exacerbate risks relative to biodiversity, human and ecosystem health, infrastructure, and food systems. To reach the stated emission reduction targets, defining the regional and temporal evolution characteristics of land-use carbon emissions and identifying their driving mechanisms, optimizing land-use structures, and transitioning to green and low-carbon development are crucial.

The Yellow River Basin is the location where human civilization began in China. It has had significant influences on agricultural output, urban expansion, economic growth, and the advancement of civilization. Since ancient times, China’s economic and cultural development have been centered around the Yellow River Basin. According to the International Energy Agency (IEA), China’s industrial production-related carbon emissions increased from 71% of total emissions in 1990 to 83% in 2018. The Yellow River Basin serves as China’s important source of oil, coal, and other energy resources and connects the Qinghai–Tibet Plateau, the Loess Plateau, and the North China Plain. The stark contrast between the basin’s economic and social development and the preservation of the natural ecosystem is apparent [8,9]. Its management also constitutes the most important and challenging aspect of river basin management [10]. Since 2019, the ecological preservation and high-quality development of the Yellow River Basin have been prioritized in a significant national policy [8]. An essential component for achieving high-quality development is green, low-carbon development.

Therefore, it is particularly crucial to measure the land-use carbon emissions index in 95 cities in the Yellow River Basin. It is variable in the economy, society, and natural environment of each city in the Yellow River basin. In order to analyze the differentiation characteristics of the spatio-temporal pattern, spatial autocorrelations were used to analyze the spatial correlations between them. Moreover, in order to monitor land-use carbon emissions as they occur in time and space, a geographically and temporally weighted regression was used to quantitatively analyze the driving mechanisms. The objective of this study is to contribute to global climate governance while offering a theoretical foundation and policy recommendations for the high-quality development of the Yellow River Basin and the achievement of China’s “dual carbon” goal.

## 2. Literature Review

Recently, scholars analyzed land-use carbon emissions from three angles: accounting, spatio-temporal changes, and the driving mechanism. In the case of the carbon emission accounting of land-use, the bookkeeping model constructed by Houghton [11] is the most used. Based on this, scholars developed carbon emission accounting methods for land-use, such as the emission inventory method [12], sample land inventory method [13], ecosystem process model [14], and carbon emission coefficient method [15]. In this field, numerous findings have been obtained [16,17,18,19]. However, due to the complexity of human activities and the uncertainty of many factors in nature, providing a quantitative description of certain land-use factors is difficult. Consequently, constructing a unified method for the calculation of land-use carbon emissions is difficult.

Initially, scholars examined the characteristics of the evolution of a study area’s spatio-temporal patterns by calculating the amount of land-use carbon emissions produced. According to the quantity of the emissions and the level of economic growth, Hong et al. [20] classified the world’s land-use carbon emission patterns into four zones. Tian et al. [21] observed a general downward trend in global carbon emissions from 1992 to 2015, which is caused by changes in land use, with the majority of the decline occurring in South Africa, Central Africa, and Southeast Asia. The majority of East Asia, Central Africa, northwest North America, and eastern South Africa experienced a rise in such emissions. According to Lin [22], China’s land-use carbon emissions increased from 2006 to 2016, with a clear north–south divergence in carbon sink sources. In conclusion, land-use carbon emissions exhibited an increasing trend over time, with construction lands serving as the primary carbon source and forests and grasslands serving as the primary carbon sinks [23,24,25]. Land-use carbon emissions vary significantly from a spatial standpoint, and a degree of spatial connection or heterogeneity is present [26,27].

What caused the formation of these spatio-temporal patterns? Scholars have become interested in the mechanism behind carbon emissions from land usage. Early research concentrated mainly on the effect of changes in land-use structure on carbon emissions, and as early as 1986, Detile [28] discovered that a change in land use might increase or decrease carbon emissions. Furthermore, Houghton [29] revealed that carbon dioxide levels on Earth increase because of deforestation and the conversion of forest areas for various uses and that the rate of carbon dioxide release is correlated with the intensity of how humans use lands. Yang et al. [30] discovered that, in China, the destruction of forest ecosystems, particularly the conversion of forests to croplands and grasslands, can result in a significant release of carbon from the terrestrial biosphere into the atmosphere, an amount that can be compared to that produced by burning fossil fuels. Additionally, scholars are interested in the connection between social and economic activity and land-use carbon emissions. Feng et al. [31] found that, in China, land-use carbon emissions are positively correlated with gross domestic product (GDP) and negatively correlated with population size. According to Wu et al. [32], high demands for energy-intensive land-use types and poor land management significantly contribute to the increase in carbon emissions in Zhejiang Province. Zhao et al. [33] discovered that Nanjing’s land usage per unit of GDP serves as a deterrent to rising carbon emissions, while other factors function as drivers.

The current literature on land-use carbon emissions focuses either on the majority of the country’s territory or on a single province. Researchers paid less attention to the basin because of the high complexity. Examining the development of land-use carbon emission patterns and driving processes is crucial given the significance of the Yellow River Basin in China and in the context of transitions to green and low-carbon development. Additionally, the majority of the current research on the causes of land-use carbon emissions concentrates on environmental and socioeconomic variables and frequently ignores the influence of policy, technology, thought, and other human elements. Therefore, a more extensive study of land-use carbon emissions needs to be performed in order to improve the public’s understanding of its causes.

## 3. Study Area and Methodology

### 3.1. Study Area

The Yellow River, one of the six longest rivers in the world and the second longest river in China, is situated in the northern regions of the country. The basin spans 795,000 square kilometers in total (including 42,000 square kilometers of flow area). In 2020, the provinces in the Yellow River Basin had an overall population of 420 million people, which was 30.3% of China’s entire population. The GDP of the region was 23.9 trillion CNY or 26.5% of the nation’s total. The GDP of the upper, middle, and lower reaches contributed 14.54%, 21.27%, and 64.19%, respectively, to the GDP of the basin as a whole. The industrial population in the Yellow River Basin is primarily concentrated in the lower reaches, with a total of seven urban agglomerations in terms of the spatial structure of social and economic development.

The Sichuan Province will not be included in this study as it is part of the Yangtze River Basin and does not form the mainstream of the Yellow River. The study area has 95 municipal administrative units and eight provincial administrative units, including Shandong, Shanxi, Henan, Shaanxi, the Ningxia Hui Autonomous Region, Inner Mongolia, Gansu, and Qinghai (Figure 1). It covers a total area of 2,995,100 square kilometers.

Given the long duration of the current study and the high variability of the administrative units, the 2020 administrative territorial entity was adopted as the standard for analytical purposes; thus, the data of each year were consolidated into the 2020 administrative units. Five periods of land-use data in the Yellow River Basin, i.e., those of 2000, 2005, 2010, 2015, and 2020, obtained from the Chinese Academy of Sciences Environmental Science Data Center with a resolution of 30 m were used as data sets in this study. Based on the LUCC (Land-use Cover Change) classification system, the data were reclassified into six categories: arable land, forestland, grassland, water area, construction land, and unused land.

### 3.2. Methodology

This research study evaluates land-use carbon emissions, identifying its spatio-temporal patterns, spatial associations, and driving mechanisms over spatial and temporal dimensions. The research methods adopted here are methods utilized for estimating land-use carbon emissions, global spatial autocorrelation, and geographically and temporally weighted regression models (Figure 2).

#### 3.2.1. Methodology for Estimating Land-Use Carbon Emissions

Direct carbon emissions and indirect carbon emissions are two different types of land-use carbon emissions. Direct carbon emissions refer to emissions from human activities performed on land, while indirect carbon emissions mostly relate to carbon emissions caused by human activity on construction sites and, more specifically, energy usage. A model estimation based on the Intergovernmental Panel on Climate Change’s (IPCC) inventories is widely used for calculating land-use carbon emissions, which typically utilizes the carbon emission coefficient [34].

The following equation can be used to calculate direct land-use carbon emissions:(1)$$E_{k}=\Sigma E_{i}=\Sigma S_{i}\times \theta_{i}$$

In the formula, $E_{k}$ is the direct land-use carbon emission and $E_{i}$ is the carbon emissions from different types of land-use. $S_{i}$ is the area of land-use type, and $\theta_{i}$ is the carbon emission coefficient of land-use type. Based on the results of relevant studies [35,36,37] and the geographical conditions of the Yellow River Basin, the carbon emission coefficients for the land types are determined. Cultivated land has the ability of both being a carbon source and permitting carbon sequestration processes. Thus, the net carbon emission coefficient of cultivated land equals the carbon emission coefficient minus the carbon absorption coefficient. The carbon sequestration of cultivated land is reflected in the cultivated land crops and soil organic carbon pools. Meanwhile, agricultural production activities will lead to carbon emissions. Via Cai’s studies, it can be concluded that the carbon emission coefficient of cultivated land is 0.0504 t/ha and the carbon absorption coefficient is 0.0007 t/ha [38]. Therefore, the net carbon emission coefficient of cultivated land was determined to be 0.0497 t/ha in this paper. Woodland has a strong carbon sequestration capacity. According to Xiao’s study [15], the weighted average of China’s forest coefficient is −0.0581 t/ha as the woodland carbon emission coefficient. In this paper, the average value was determined as the carbon emission coefficient of forest land. Grassland is mainly used for carbon sequestration via photosynthesis. According to Shi’s study [35], the carbon emission coefficient of grassland in the Yellow River Basin was determined to be −0.021 t/ha. Water is an essential component of the ecosystem and performs a variety of ecosystem services for the local environment, including soil carbon sequestration. By examining the available data and integrating the data with the actual situation observed in the water region, Wang determined that the average carbon sink coefficient of the Yellow River Basin is 0.0041 t/ha [39]. Therefore, the carbon emission coefficient of the water was determined to be −0.041 t/ha. Unused land has a relatively low secondary terrestrial carbon absorption capacity in the Yellow River Basin. The carbon emission coefficient of unused land was determined to be generally −0.0005 t/ha [40].

The following equation can be used to calculate indirect land-use carbon emissions.
(2)$$E_{t}=\Sigma E_{j}=\Sigma C_{j}\times \alpha_{j}\times \beta_{j}$$

In the formula, $E_{t}$ is the indirect carbon emission of land-use and $E_{j}$ is the carbon emission of the type of statistical energy consumption. $C_{j}$ is the amount of energy consumed $\alpha_{j}$ is the standard coal conversion coefficient for various energy sources, and $\beta_{j}$ is the carbon emission coefficient for various energy sources. This study analyzes eight main energy sources: coal, coke, crude oil, diesel oil, kerosene, gasoline, electric power, and natural gas. The coal conversion indices and carbon emission coefficients for different energy standards were obtained from the *China Energy Statistical Yearbook* and the IPCC guidelines for national greenhouse gas inventories [41] (see Table 1).

#### 3.2.2. Global Spatial Autocorrelation

Global spatial autocorrelation was used to measure the spatial concentration of land-use carbon emissions in the cities of the Yellow River Basin. Global spatial autocorrelation is a description of the spatial properties of the attributed values for the entire region [42]. Moran’s I index, which is most frequently used, was determined using the following formula.
(3)$$I=\left(\frac{n}{\sum_{i=1}^{n}\sum_{j=1}^{n}W_{\mathrm{ij}}}\right)\left(\frac{W_{\mathrm{ij}}\left(x_{i}-\;\bar{x}\right)\left(x_{j}-\bar{x}\right)}{\sum_{i=1}^{n}{\left(x_{i}-\bar{x}\right)}^{2}}\right)$$

In the formula, I is the global Moran value, n is the number of cities, $x_{i}$ and $x_{j}$ are the observations of land-use carbon emission, and $\bar{x}$ is the average of $x_{i}$. $W_{ij}$ is the spatial weight matrix. When entity i is topologically adjacent to entity j with a common edge, the value is 1; otherwise, it is 0.

#### 3.2.3. Local Spatial Autocorrelation

To assess the characteristics of spatial clustering in the study’s region and identify the high- and low-value clustering areas, local spatial autocorrelations were used to calculate the value of the local autocorrelation statistics for each spatial unit [43]. In this study, the local autocorrelation index, local indicators of spatial association (LISA), was employed to determine the different clustering types of local spatial units. The equation used is as follows.
(4)$$I_{i}=Z_{i}\sum_{i=1}^{n}W_{\mathrm{ij}\mathrm{Zj}}$$

In the formula, $I_{i}$ is the local spatial autocorrelation index, $W_{ij}$ is the spatial weight matrix, and $Z_{i}$ and $Z_{j}$ are the standardized values of the land-use carbon emissions. According to this, the research units can be divided into four spatial correlation patterns: High–High (H-H), Low–Low (L-L), High–Low (H-L), and Low–High (L-H).

#### 3.2.4. Geographically and Temporally Weighted Regression Model

In the current literature on spatial heterogeneity, the geographically weighted regression (GWR) model is frequently utilized, although it uses cross-sectional data, which is prone to anomalous data fluctuations and parameter overshoots. However, including the time dimension, i.e., by using the geographically and temporally weighted regression (GTWR) model, helps overcome the aforementioned issues and improve the accuracy of the estimation findings [44]. Thus, the GTWR model was used in this study to examine the spatio-temporal variations of numerous contributing factors. The formula of the GTWR model is as follows.
(5)$$y_{i}=\beta_{0}(u_{i},v_{i},t_{i})+\sum_{k=1}^{p}\beta_{k}(u_{i},v_{i},t_{i})X_{ik}+\sum_{k=1}^{p}\beta_{k}(u_{i},v_{i},t_{i})WX_{ik}+\varepsilon_{i}$$

In the formula, $y_{i}$ represents the land-use carbon emissions in the Yellow River Basin, W is the spatial weight matrix, $X_{ik}$ is the vector of an exogenous explanatory variable, $\beta_{k}$ is the coefficient of the *k*th variable, which reflects the influence of the *k*th independent variable $X_{k}$ on the dependent variable, $\varepsilon_{i}$ is the error term, and $\left(u_{i},v_{i},t_{i}\right)$ represents the spatio-temporal location of city i in year t.

Population size (*P*), economic level (*PGRP*), land-use structure (*LUS*), industrial structure (*IS*), social development (*UR*), and technological progress (*TP*) were selected as independent variables for constructing the index system. Among them, the population scale is expressed as the population density, the economic level is expressed by per capita regional GDP, and the land-use structure is expressed by the proportion of construction land to total land usage. Moreover, the industrial structure is expressed in terms of the share of the secondary sector of the economy in the GDP, social development in terms of the rate of urbanization, and technological progress in terms of the number of researchers. The data were mainly obtained from the *China Urban Statistical Yearbook* and every city’s statistical yearbook from 2000 to 2020. Some missing data were filled in by using the interpolation method.

## 4. Results

### 4.1. Spatio-Temporal Patterns of Land-Use Carbon Emissions in the Yellow River Basin

The land-use carbon emissions from 2000 to 2020 of 95 prefecture-level cities in the Yellow River Basin were calculated (Table 2). The findings revealed that since 2000, carbon emissions have been on the rise in the Yellow River Basin. The net carbon emissions rose from 35,395.67 × 10^4^ t in 2000 to 141,837.37 × 10^4^ t in 2020. According to the *China Urban Statistical Yearbook* from 2000 to 2020, the per capita GDP increased by about 51,945.78 CNY, the secondary industry output value increased by about 40 times, and the urbanization rate increased from 28.58 percent to 57.68 percent in the Yellow River Basin. The rapid industrialization, urbanization, and economic growth could have an impact on the rise in net carbon emissions in the Yellow River Basin. Carbon emissions increased quickly between 2000 and 2010, with the growth rate peaking between 2005 and 2010 and then progressively declining after 2010. The fundamental cause of the sluggish growth in carbon emissions is the gradual rationalization of the structure of land-use, industry, and energy [45]. Arable and construction land supply the majority of the Yellow River Basin’s carbon emissions, with construction land serving as the primary source. Indeed, carbon emissions from construction land, which account for more than 92% of the total carbon emissions, exhibited a sharp increase in the previous two decades from 35,193.31 × 10^4^ t to 141,764.52 × 10^4^ t. In the last 20 years, carbon emissions from arable land decreased slightly from 2734.51 × 10^4^ t to 2645.95 × 10^4^ t, which exhibits a fairly consistent trend. The main carbon sinks are forestland, grassland, water area, and unused land, with forestland accounting for more than 82% of the total sinks for carbon emissions; other land-like carbon sinks are weak. During the study’s period, there was little change in the carbon emissions of these four types of land.

Although they have become stable in recent years, overall carbon emissions in the Yellow River Basin are still rising increasing and have not yet attained their peak. Additionally, a significant difference exists between the carbon sink and the source over the same period, indicating that carbon absorption from land use alone is insufficient to offset the high carbon emissions caused by excessive energy consumption. Thus, the emission reduction policy should continue to prioritize improving energy and industrial structures, technological innovations, and emission reductions.

The ArcGIS software was used to visualize the data on land-use carbon emissions in the study area and analyze the spatio-temporal patterns of the carbon emissions in the Yellow River Basin (Figure 3). Land-use carbon emissions were graded on a six-point scale, from low to high, to show the extent of carbon emissions globally.

The Yellow River Basin as a whole experienced low carbon emission levels in 2000. By 2005, Taiyuan, Xi’an, Qingdao, and Yantai exhibited greatly increased carbon emissions, which remained at a high level from this year onward. Between 2005 and 2015, the number of cities with high and very high carbon emission levels rose to 18, with significant increases in the carbon emissions of Ordos, Hohhot, and Baotou. By 2020, the Yellow River Basin’s high carbon emission areas mostly remain unchanged. Carbon emissions in most cities are expected to climb, although they will decline in some cities. This is the result of China’s focus on low-carbon growth, energy conservation, and emission reduction and the promotion of upgrades relative to industrial energy structures in recent years. Evidently, the middle reaches of the Yellow River Basin have a higher growth rate with resepct to carbon emissions than the upper and lower reaches. This is because the middle reaches of the Yellow River Basin are significant energy and heavy industrial development zones that are highly dependent on energy production and energy consumption. Consequently, the middle reaches experience significant increases in carbon emissions due to undergoing rapid industrialization and urbanization. Over a period of 20 years, land-use carbon emissions in Xi’an rose the most, rising by 6160.99 × 10^4^ t from 977.52 × 10^4^ t in 2000 to 7138.51 × 10^4^ t in 2020. Most of the top 10 cities are located in the Inner Mongolia, Shanxi, Shaanxi, and Shandong provinces, which points to the severity and breadth of the need for emission reductions in the middle and lower regions of the Yellow River Basin. The provinces of Qinghai and Gansu had cities with the least amount of increases in land-use carbon emissions, with only a 36.71 × 10^4^ t increase in carbon emissions in the Golog Tibetan Autonomous Prefecture between 2000 and 2020. This is because the upstream region’s urbanization process is relatively slow and the region is less dependent on energy consumption.

Overall, the Yellow River Basin’s land-use carbon emissions increased significantly from 2000 to 2020. The spatial pattern of “high in the middle east and low in the northwest” is evident; the regions with the highest emissions are primarily the Inner Mongolia, Shanxi, Shaanxi, and Shandong provinces, while the regions with the lowest emissions are primarily the Gansu and Qinghai provinces. Cities in the middle and lower reaches of the Yellow River Basin play an important role in food security, energy supply, and industrial development. Rapid economic development, urbanization, and industrialization cannot be separated from the large amount of energy consumption and the conversion of a large amount of land to construction land. In hindsight, this method was the only option at the time for China’s contemporary growth, but the numerous environmental problems it gave rise to still require attention. To reach the aim of “peak carbon by 2030, carbon neutrality by 2060,” China needs to implement policy measures more swiftly and effectively, which would help optimize its energy structure, industrial structure, and land-use structure.

### 4.2. Spatial Correlation Analysis of Land-Use Carbon Emissions in the Yellow River Basin

Using carbon emissions data from 95 cities as samples, global autocorrelation (Table 3) and local autocorrelation analyses (Figure 4) were performed to further investigate the spatial distribution of land-use carbon emissions in the Yellow River Basin. The research units can be divided into four spatial correlation patterns: High–High (H-H), Low–Low (L-L), High–Low (H-L), and Low–High (L-H).

According to Table 3, a significant spatial correlation of land-use carbon emissions is present in the study’s area, with Moran’s I values of land-use carbon emissions in the Yellow River Basin exceeding 0.1 in 2000, 2005, 2010, 2015, and 2020 and confidence levels of Z values in each year exceeding 99%. Moran’s I value increased between 2000 and 2005, demonstrating that carbon emissions in the Yellow River Basin had a significant regional spillover effect during this time. From 2005 to 2020, Moran’s I value gradually fell, indicating a decrease in the geographical concentration of cities with similar land-use carbon emissions. However, the overall land-use carbon emissions in the Yellow River basin are still in an obvious state of agglomeration, and the geographical differentiation features are slowly becoming prominent.

Despite reflecting the spatial correlation between land-use and carbon emissions of various cities in the Yellow River Basin, the global spatial autocorrelation index is unable to understand the precise spatial relationships between various cities. To examine the substantial local spatial correlation of land-use carbon emissions in various cities in the Yellow River Basin, this study employed local spatial autocorrelations.

According to Figure 4, “H-H” and “L-L” were the most common spatial clustering types of land-use carbon emissions in the Yellow River Basin, and “H-L” and “L-H” were not significant. In 2000, the “H-H” agglomeration areas were spread over the provinces of Inner Mongolia, Shanxi, and Henan. From 2005, the Shandong province started to exhibit a significant positive agglomeration impact, while the Henan province steadily withdrew from the “H-H” agglomeration area. The “L-L” cluster is centered in Qinghai Province, to which Zhangye belonged to in 2005, 2010, and 2015. China’s Qinghai Province has long served as a significant ecological security barrier. It neither depends heavily on energy consumption nor carries out rapid land urbanization expansion during rapid economic development. Thus, the total land-use carbon emissions of Qinghai is minimal, and it is relatively less affected by the changes and restrictions imposed on other provinces. The spatial dependence of various cities in the province is relatively stable. Generally, the “H-H” and “L-L” concentrations of land-use carbon emissions were mostly concentrated in the middle and lower sections of the Yellow River Basin, while the “L-L” concentrations were mostly concentrated in the upper reaches. This phenomenon illustrates the significant spatial agglomeration effect of land-use carbon emissions in the Yellow River Basin and reaffirms the spatial pattern of total carbon emissions of “high in the middle east and low in the northwest”.

### 4.3. Analysis of the Driving Factors of Carbon Emissions from Land Use in the Yellow River Basin

Both internal and external factors contribute to the evolution of the spatio-temporal pattern of land-use carbon emissions. In addition to intrinsic land-related elements, social and economic factors also affect land-use carbon emissions. The GTWR model was used to assess the six driving elements of this study: population, economy, society, land, industry, and technology. The *AICc* was 2887.05 with a bandwidth of 0.1141, showing that the model’s capacity for interpretation is strong (Table 4).

By creating box diagrams (Figure 5), the spatio-temporal variation of each driving factor’s coefficient was observed, and its shifting trend and cause were examined.

Land-use carbon emissions are influenced by the population size of the region. In 2000–2005, the driving effect of population size on the cities was mainly negative inhibition; that is, the increase in population size generally brought about a reduction in net land-use carbon emissions, and the impact coefficient was concentrated within −1.68 and 0.69. However, the driving effect of the population size progressively turned into a positive incentive effect from 2010, the impact coefficient generally varied between 0 and 4, and the influence coefficient of many cities shifted from negative to positive. Observations revealed that in the early stages, the areas with a positive population scale influence coefficient were mostly concentrated in the upper reaches of the Yellow River Basin. Due to the delayed urbanization process and inadequate technological infrastructure in these cities, it is difficult to turn the increase in population size into a driving force. The effect coefficient of the region’s lower and middle reaches gradually increased over time. This is because as China’s population grows, the urban population’s redundancy and high energy use further increase carbon emissions.

Land-use carbon emissions are driven by economic development. Between 2000 and 2020, there was a negative trend in the impact coefficient of economic development in the cities in the Yellow River Basin, which ranged from 0 to 8. This demonstrates that the amount of economic growth in the area had a clear but progressively diminishing positive driving effect on land-use carbon emissions. The influence coefficient was high in the middle and lower reaches and low in the upper reaches. This is due to the excessive attention that was given to the rapid speed of economic growth in the early development of the Yellow River Basin while ignoring its quality and sustainability, which led to cities pursuing economic development by overusing local resources and untapped development potential. In particular, the middle and lower reaches of the Yellow River have become overly dependent on the growth of heavy industry and energy consumption, which has led to a “high economy, high energy consumption, and high pollution” development model. Thus, the level of economic development has a significant impact on land-use net carbon emissions. Conversely, the upper reaches’ level of economic development and its reliance on energy are both lower. With the promotion of green and low-carbon development policies in recent years, cities optimized their economic and energy structures and attempted to realize the “decoupling” of land-use carbon emissions and economic development levels, leading to a decline in the influence coefficient of economic development levels. Some cities even have negative impact coefficients, which show that local governments are allocating more of their GDP to the use of clean energy and the study and development of energy-saving and emission-reducing technologies. Consequently, the level of economic development has had a negative inhibitory effect on land-use carbon emissions. However, it cannot be denied that the Yellow River Basin and China as a whole are still experiencing economic growth. Mindlessly sacrificing economic development in favor of carbon emission reduction would be unscientific, as attempts to reduce carbon emissions and increase economic growth can occur simultaneously. Therefore, governance needs to be more precise and scientific to achieve the coordinated economic development and ecological conservation.

Land-use carbon emissions are influenced by the region’s industrial structure. The growth in the share of the secondary industry had a favorable impact on net land-use carbon emissions in the Yellow River Basin in 2000, according to the impact coefficient of industrial structure, which was positive and ranged from 0 to 1. However, starting in 2005, the industrial structure’s effect coefficient started to decline. By 2020, it mostly fluctuated from −4 to 0.2, and the degree of dispersion deepened over time. It is worth noting that in the cities with a lower level of development in the upper reaches of the Yellow River Basin, such as the Golog, Gannan, and Huangnan Tibetan Autonomous Prefectures, the influence coefficient of industrial structures is still large. However, in the middle and lower reaches of the Yellow River, the influence coefficient of industrial structures is small and mostly negative. This further demonstrates that the lower the degree of development, the more imperfect the technology; the more energy-intensive the city, the higher the city’s carbon emission, and the closer the industrial structure.

Land-use carbon emissions are driven by land-use structures at the social development level. In 2000, the influence coefficient of social development levels on cities in the Yellow River Basin was concentrated within −0.5 and 0, and the spatial difference was small, indicating that population urbanization had a negative inhibitory effect on the net carbon emissions of land use. The influence coefficient of the level of social development climbed steadily between 2005 and 2020, fluctuating between 0 and 4, and the spatial disparity grew. From 2000 to 2020, as the difference between the urban development levels in the Yellow River Basin widened, the range of the influence coefficient of social development levels increased continuously, reaching 12.68 in 2020. Thus, while reducing carbon emissions, the focus should be more on coordinated regional developments.

Land-use carbon emissions are influenced by technological progress. From 2000 to 2020, the Yellow River Basin’s land-use carbon emissions were significantly positively encouraged by technological advancement, and the impact coefficient varied within the range of 3 to 8, with the spatial difference first increasing and then reducing. The upper sections of the Yellow River Basin are home to most cities that have a negative impact coefficient of technological advancement from a spatial distribution standpoint. This is because there is less pressure for economic development in the upper reaches of the river region than in the middle and lower reaches, where the priority is ecological protection. Therefore, earlier technological advancements that were biased toward energy saving and emission reduction had negative inhibitory effects on carbon emissions from land use. Several initiatives to reduce carbon emissions in China have been successful in recent years, as evidenced by the decreased impact coefficient of technological progress in cities in the middle and lower reaches of the Yellow River Basin since 2015.

Land-use carbon emissions are driven by land-use structures. In 2000, no regional variation was present in the impact coefficient of land-use structures in cities in the Yellow River Basin, which ranged between 0 and 2. Since 2005, more cities emerged with negative effect coefficients, a progressive deepening of the dispersion of the impact coefficient, and a notable increase in regional disparity. Additionally, the growth of construction land decreased due to macroeconomic regulations and control by the Chinese government, which have cumulatively led to a decrease in net land-use carbon emissions. Moreover, the lower and lower-middle reaches of the Yellow River Basin are home to the majority of cities with a positive effect coefficient, while the upper and upper-middle reaches are home to the majority of cities with a negative impact coefficient. This demonstrates that the vast land use in downstream cities such as Zhengzhou and Qingdao continues to lead to a wastage of resources and energy, which increases net land-use carbon emissions.

## 5. Discussion

The current studies on land-use carbon emissions focus either on the majority of the country’s territory or on a single province, with basin systems receiving less attention. The Yellow River Basin is China’s primary energy source and residence to a sizable population. The study of the spatial-temporal pattern of land-use carbon emissions in the Yellow River Basin is crucial. Furthermore, few studies have combined the two dimensions of time and space to examine the influencing factors of land-use carbon emissions. Considering the spatial heterogeneity of the Yellow River Basin, the GTWR model was chosen in this paper to identify the influencing factors and to render the research results more scientific.

According to this study’s findings, which are consistent with those of earlier studies [22,46], construction land is the primary carbon source, whereas forest lands provide the strongest carbon sequestration processes. This study discovers that net carbon emissions in the Yellow River Basin have an obvious spatial link and that the spatial distribution follows a southeast-to-northwest pattern, which is in accordance with the opinions of academics such as Ma Yuan [47]. Some researchers investigated the elements that affect land-use carbon emissions and discovered that the population size had a beneficial impact on emissions with a modest increase [48,49]. We examined the change in the driving factor coefficients from the time and space dimensions by the GTWR model. Different outcomes were obtained. It was discovered that the effect of driving factors on land-use carbon emissions is not always consistent. Population size is one example of a variable that can have both positive and negative effects.

There are some limitations to this study that can be improved in the future. First, no standard for the land-use carbon emissions index currently exists. Future studies will concentrate on determining the best way to evaluate the land-use carbon emissions index in relation to different spatial scales and areas. Second, this paper selected many factors that influence land-use carbon emissions, including population size, economic development level, industrial structure, social development level, technological advancement, and land-use structure. However, policy control has a significant impact on the sustainable development of the national economy and environment. Therefore, the focus of future studies will primarily be on policy control.

## 6. Conclusions

### 6.1. Main Conclusions

The temporal and spatial patterns were determined by measuring the land-use carbon emissions of 95 cities in the Yellow River watershed from 2000 to 2020. The GTWR model was then used to investigate the causes of land-use-related carbon emissions in the region. The following primary conclusions were obtained:(1)The total net land-use carbon emissions in the Yellow River Basin dramatically changed between 2000 and 2020, with the growth rate first rising and then falling. The primary carbon sink is forest land, while the primary carbon source is construction land.(2)Over the past 20 years, the net land-use carbon emissions of cities in the Yellow River Basin increased to varying degrees, with the largest increase in Xi’an and the smallest increase in the Golog Tibetan Autonomous Prefecture. The spatial pattern of land-use carbon emissions is “high in the middle east and low in the northwest,” which is closely related to urban development models and function positioning.(3)The Yellow River Basin’s urban land-use carbon emissions showed a substantial spatial link, with the degree of spatial agglomeration initially rising and then falling. “L-L” was concentrated in the upper parts of the Yellow River Basin and was generally stable, while “H-H” was concentrated in the middle and lower reaches with an eastward migration trend.(4)According to the analysis of the GTWR model, the Yellow River Basin’s land-use carbon emissions are driven by a variety of factors, including population size, economic development level, industrial structure, social development level, technological advancement, and land-use structure. While the rates of economic development and technological advancement have a major positive driving effect, the other components’ driving effects fluctuate with time, and significant geographical variances exist.

### 6.2. Policy Implications

The following policy recommendations are made to ensure the achievement of the “dual carbon” target:(1)Controlling the size of the population and advance the quality of the population: The influence of population density on land-use carbon emissions is known. More than 30% of China’s entire population is in the Yellow River Basin, which is a sizable percentage. Along with the implementation of the universal three-child policy, population growth will drive carbon emissions in the short term. To avoid rapid population growth following the implementation of the three-child policy, people must be guided toward a correct conception of fertility. Additionally, it is important to keep the inflow of outlanders under control and to strike a balance between reducing carbon emissions and luring outlanders to create jobs. The higher quality of the population will inevitably advance technology and maximize the benefits of human capital.(2)Learning advanced carbon sequestration techniques and improving the function of carbon sequestration: The Chinese Ministry of Agriculture and Rural Affairs published 10 technology models for carbon mitigation and sequestration in 2021. These models cover important topics such as carbon mitigation and sequestration in the planting industry, rural renewable resource replacement, and straw-return processes, which provide two advantages: stable production and supply and carbon mitigation and sequestration. The government should invest more funds in advanced machinery. Moreover, studying advanced science and technology should be encouraged.(3)Strengthening regional cooperation and promoting coordinated development among regions: The Yellow River Basin’s urban land-use carbon emissions showed a substantial spatial link. Therefore, the overall emission reduction target can be met by implementing regionally collaborative emission reduction policies. The inter-regional economic cooperation linkage and compensation mechanism should be established to promote the efficient circulation of innovative resources in the upper parts and the middle and lower parts of the Yellow River Basin.(4)Pay attention to regional variations and adapt actions to local circumstances. The volume and cause of land-use carbon emissions in the Yellow River’s watershed varied significantly by location. Each province should leverage its geographic, cultural, and natural advantages to expand its market. For example, Shandong and Shanxi can rely on the natural and cultural landscapes to develop tourism vigorously, and Henan can use its national transportation hub to develop the economy vigorously.

## Figures and Tables

**Figure 1 ijerph-19-16507-f001:**
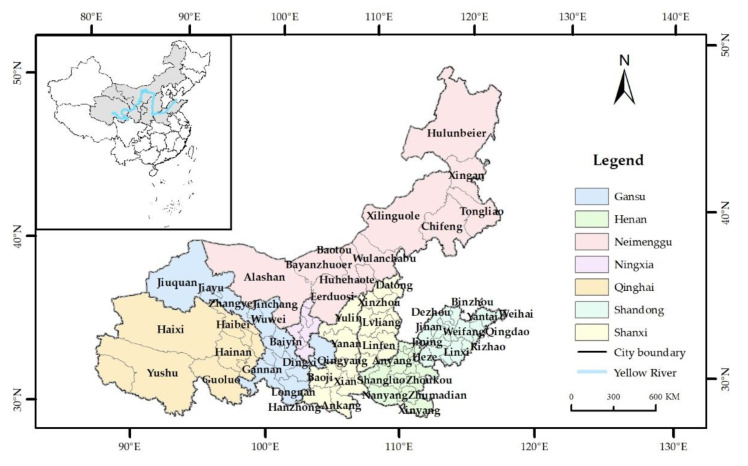
Location of the study area.

**Figure 2 ijerph-19-16507-f002:**
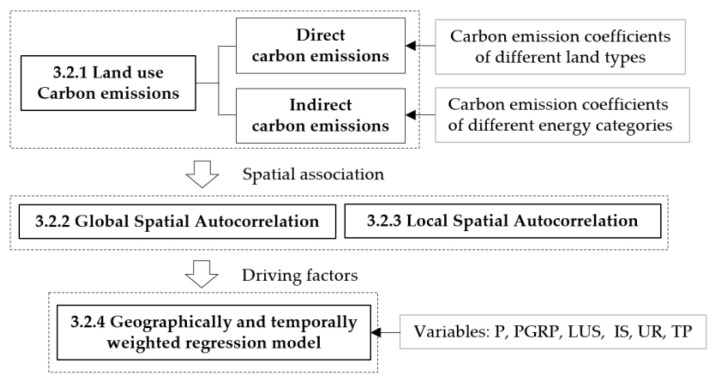
Flow diagram of the methodology.

**Figure 3 ijerph-19-16507-f003:**
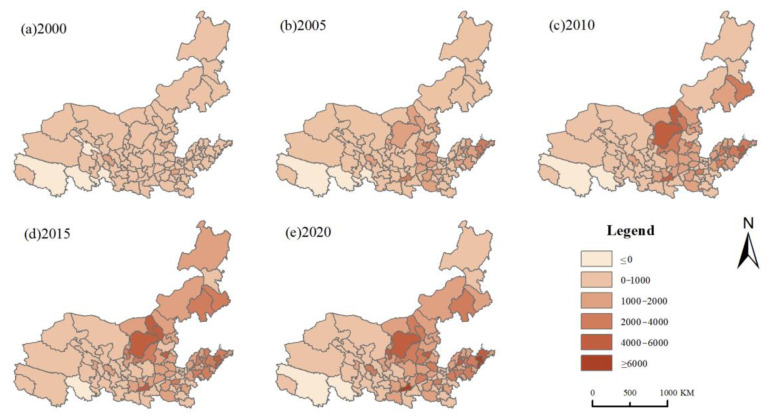
Land-use carbon emissions of cities in the Yellow River Basin from 2000 to 2020 (10^4^ t).

**Figure 4 ijerph-19-16507-f004:**
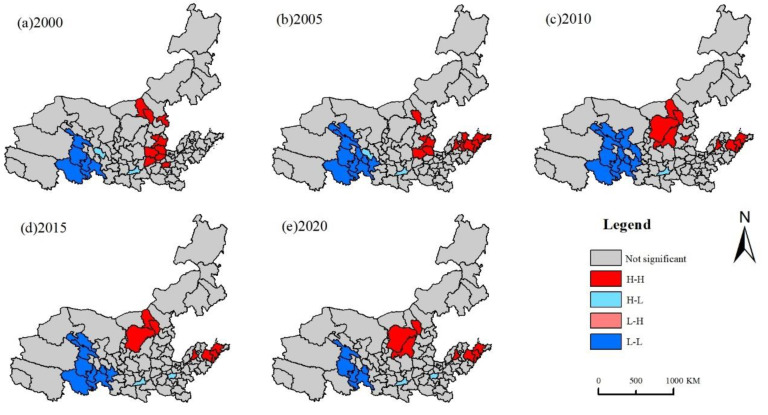
LISA spatial agglomeration pattern of land-use carbon emissions in the Yellow River Basin from 2000 to 2020.

**Figure 5 ijerph-19-16507-f005:**
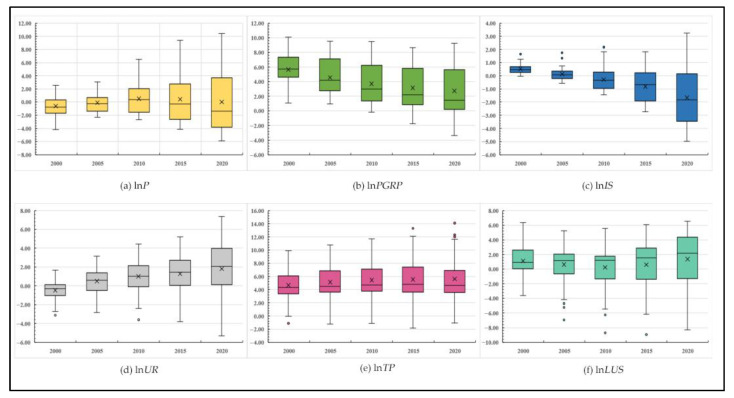
Driving mechanism coefficients of the land-use carbon emissions in the Yellow River Basin.

**Table 1 ijerph-19-16507-t001:** Standard coal conversion coefficients and carbon emission coefficients of different energy categories.

Energy Category	Standard Coal Conversion Coefficients	Carbon Emission Coefficients (t/t)
Coal	0.7143 kg/kg	0.7559
Coke	0.7914 kg/kg	0.8550
Crude oil	1.4286 kg/kg	0.5857
Diesel oil	1.4571 kg/kg	0.5921
Kerosene	1.4714 kg/kg	0.5714
Gasoline	1.4714 kg/kg	0.5538
Electricity	0.4040 kg/kg	0.7935
Natural gas	1.2143 kg/m^3^	0.4483

**Table 2 ijerph-19-16507-t002:** Carbon emissions of different land types in the Yellow River Basin from 2000 to 2020 (10^4^ t).

Year	Cultivated Land	Woodland	Grassland	Water	Construction Land	Unused Land	Net Carbon Emissions	Growth Rate
2000	2734.51	−2092.59	−254.81	−147.25	35,193.31	−37.50	35,395.67	—
2005	2708.57	−2111.63	−253.97	−150.29	68,014.45	−37.57	68,169.56	93.42%
2010	2680.78	−2118.27	−255.38	−156.16	114,601.18	−37.15	114,715.65	68.80%
2015	2672.00	−2117.54	−256.37	−159.52	133,271.17	−36.47	133,373.27	16.09%
2020	2645.95	−2111.24	−257.15	−168.68	141,764.52	−36.03	141,837.37	6.05%

**Table 3 ijerph-19-16507-t003:** Global Moran’s I of land-use carbon emissions in the Yellow River Basin from 2000 to 2020.

Year	2000	2005	2010	2015	2020
Moran’s I	0.1196	0.1860	0.1633	0.1399	0.1113
Z(*I*)	3.8113	5.7415	5.0785	4.4022	3.5943

Note. Z(*I*) > 1.96 indicates a significance level of 5%, and >2.58 indicates a significance level of 1%.

**Table 4 ijerph-19-16507-t004:** The results of the GTWR.

Dependent Variable	Bandwidth	Res.^2^	Sigma	AICc	R^2^	Adjusted R^2^	STDR	Trace of Matrix
CE_net_	0.11	8501.03	4.23	2887.05	0.86	0.85	0.27	70.18

Note: *CE_net_* represents net land-use carbon emissions, *Res.*^2^ represents the square of residuals, and *STDR* represents the spatio-temporal distance ratio.

## Data Availability

Not applicable.

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
