# Peer review of "Land-Use Carbon Emissions in the Yellow River Basin from 2000 to 2020: Spatio-Temporal Patterns and Driving Mechanisms"

_ijerph, 2022, doi:10.3390/ijerph192416507_

Round 1

Reviewer 1 Report (New Reviewer)

1、The author uses the coefficient method to calculate the carbon emissions of land use. The determination of coefficient is extremely important, but the author's description of this part is very simple and needs to be supplemented.

2、The text description and unit of the legend in the figure need to be added.

3、The author did not explain what kind of weight was used in the spatial autocorrelation part

4、Why does the author neglect the discussion part, and how to analyze the particularity of the Yellow River basin if not compared with the existing research? It is suggested to add a discussion section.

5、The results of geographically weighted regression still need to be presented in tabular form.

6、The research conclusions do not match the relevant policies, and the following problems exist:

(1). Although population size is an important factor affecting carbon emissions, China is currently facing the problem of population structure adjustment. It has become a consensus to liberalize birth restrictions. Why should the author control population size.

(2). The second suggestion is not specific to the specific situation of the Yellow River basin, and the suggestions put forward by the author are universally applicable.

(3). After discovering the spatial correlation of carbon emissions, the author should have focused on regional cooperation to control carbon emissions, but ignored this point.

(4). Different regions need to adopt different policies. It should specify which regions adopt what policies.

Author Response

Dear Reviewer,

Thank you for your comments concerning our manuscript entitled “Land-use carbon emissions in the Yellow River Basin from 2000 to 2020: Spatio-temporal patterns and driving mechanisms”. We much appreciate your considered advice and positive views of the work. We have revised the article and responded fully to all of your comments. We have highlighted the changes within the manuscript by using the "Track Changes" function in Microsoft Word. A detailed list of corrections and modifications is given below and are highlighted in the manuscript.

We hope that you will find the revised manuscript suitable for publication.

Best regards.

Reviewer 2 Report (New Reviewer)

General comments:

In this manuscript, the land-use carbon emissions from 2000 to 2020 of 95 cities in the Yellow River Basin were computed, and its characteristics of the spatial-temporal evolution and driving mechanisms were examined. This manuscript is well organized and written. However, several questions should be carefully addressed. Please see below for the comments, which may be helpful for improving the quality of the manuscript.

1. I noted that the authors mention that the manuscript uses global spatial autocorrelation, local spatial autocorrelation and geographically and temporally weighted regression model and that it is necessary to describe these approaches in the introduction to illustrate its advantages.

 2. The fourth section of this manuscript is "Results and Discussion" and it is suggested that the results and discussion be described separately. The fifth section is set to "Discussion" and lists sufficient evidence to support the results obtained.

 3. I noticed that the references cited in the manuscript were published over a long period of time. In my opinion, please cite as many references as possible from recent years in order to trace the research hotspots.

 Specific comments:

1. Lines 54-55: More than two references should be listed after “Numerous studies”.

 2. Lines 100-104: There is a lack of reference literature to support this conclusion.

 3. To make the article more readable, I suggest streamlining the "Study Area" section.

 4. Line 181: “global spatial autocorrelation” has appeared twice in a row, please check.

 5. In the "Methods" section, please cite additional references as the basis for your choice of method. For example: lines 214-215.

6. Please unify the layout of Figure 5.

Author Response

Dear Reviewer,

Thank you for your comments concerning our manuscript entitled “Land-use carbon emissions in the Yellow River Basin from 2000 to 2020: Spatio-temporal patterns and driving mechanisms”. We much appreciate your considered advice and positive views of the work. We have revised the article and responded fully to all of your comments. We have highlighted the changes within the manuscript by using the "Track Changes" function in Microsoft Word. A detailed list of corrections and modifications is given below and are highlighted in the manuscript.

We hope that you will find the revised manuscript suitable for publication.

Best regards.

Reviewer 3 Report (New Reviewer)

The manuscript aims at analyzing the Land-use carbon emissions in the Yellow River Basin, which is an interesting and significant topic. The following aspects are brought to the attention of the authors.

1.         Abstract

Lines 27-30: It is suggested that the conclusions in the summary should identify the main factors that affect the large regional differences in land use carbon emissions in the Yellow River basin.

2.         Introduction

Lines52-56

It is suggested to supplement and clarify the specific relationship and interaction between land use and climate and environmental change.

3.         Lines66, 70100-104

Need to supplement references

4. Figure 1

Line176

It is recommended to add longitude and latitude. The name of the region is unclear

5. Methodology

Line181: Duplicate "global spatial autocorrelation"

6. References shall be supplemented for the method part 3.3.2.,3.3.3,3.3.4

7. In line 247, each indicator is represented by only one data. Is it comprehensive.

8.Results and Discussions

Line260

Net carbon emissions in the table is the sum of 95 precision level cities' emissions?

Line265

It is not clear, and it is recommended to supplement specific data of carbon emission growth rate in tables or documents.

Lines266-268

Please recheck this sentence. What is the basis of this conclusion? Can the stage and quality of economic development be characterized only by net carbon emissions?

Line269

Need to supplement references for the sectence:The fundamental cause of the sluggish growth in carbon emissions is the gradual rationalization of the structure of land use, industry, and energy

The city name mentioned in Results and Discussions is not shown in the figure, which affects reading.

The results and discussion are explained together, and the limitations of the study are not clear.

9. Section 4.1 repeatedly mentions the impact of urbanization and industrialization on carbon emissions. What is the basis for judging the speed of urbanization and industrialization? Specific data support or references shall be provided.

Author Response

Dear Reviewer,

Thank you for your comments concerning our manuscript entitled “Land-use carbon emissions in the Yellow River Basin from 2000 to 2020: Spatio-temporal patterns and driving mechanisms”. We much appreciate your considered advice and positive views of the work. We have revised the article and responded fully to all of your comments. We have highlighted the changes within the manuscript by using the "Track Changes" function in Microsoft Word. A detailed list of corrections and modifications is given below and are highlighted in the manuscript.

We hope that you will find the revised manuscript suitable for publication.

Best regards.

Round 2

Reviewer 1 Report (New Reviewer)

The author has revised the existing problems. After improving the format of the paper, it can be published.

Author Response

Dear Reviewer,

Thank you for your recognition of our manuscript entitled “Land-use carbon emissions in the Yellow River Basin from 2000 to 2020: Spatio-temporal patterns and driving mechanisms”. We much appreciate your considered advice and positive views of the work. It has played an important role in improving the quality of our articles.

We have further revised the manuscript according to your suggestions, hoping to meet the publication requirements.

We hope that you will find the revised manuscript suitable for publication.

Best regards.

Reviewer 3 Report (New Reviewer)

Authors have incorporated all the comments. It is suggested to add the line of changes in the reply comments. At present, it is difficult to locate the modified content. There is a need to revise a complete paper in terms of Grammarly. However, results are fine. The structure of the paper is satisfactory and can be accepted after revision suggested.

Author Response

Dear Reviewer,

Thank you for your recognition of our manuscript entitled “Land-use carbon emissions in the Yellow River Basin from 2000 to 2020: Spatio-temporal patterns and driving mechanisms”. We much appreciate your considered advice and positive views of the work.

We apologize for causing you any inconvenience by not clearly marking the modified content previously.We have further revised the manuscript according to your suggestions, hoping to meet the publication requirements.

We hope that you will find the revised manuscript suitable for publication.

Best regards.

This manuscript is a resubmission of an earlier submission. The following is a list of the peer review reports and author responses from that submission.

Round 1

Reviewer 1 Report

Land use carbon emissions in the Yellow River Basin from 2000 2 to 2020: Spatio-temporal patterns and driving mechanisms

This study addresses carbon emissions from various developmental activities from 95 cities that are emerging within the watershed of the Yellow River Basin of China in the context of global climate change. The authors have inferred that carbon emission has increased between 2000 and 2020. However, the emission is not uniform across the basin indicating that an increase in carbon emission is associated with various developmental activities in urbanized region whereas carbon emission reduction is associated with the conservation of forest vegetation and natural serenity. For examples, the authors have observed high increase carbon emission in the middle eastern region where urbanization is rapid, whereas carbon emission is low in the northern region where the developmental activities are relatively at low intensities. Using a Geographical and Temporal Weighted Regression (GTWR) model, the authors have observed high to high emission in intense constructional areas such as in Xian, a sub-provincial metropolitan city and low to low emission in densely forested and thinly populated areas such as Tibetan Autonomous Prefecture. Based on their findings, the authors have made some policy recommendations.  The authors also have concluded that highly agglomerated cities have high carbon emission. The high agglomeration is associated with population concentration, level of economic development, industrial development, and social development.

The policy recommendations include:

1.     Since over 30% of the entire China’s population lives along the Yellow River Basin, it is necessary to regulate population growth in this region. The authors have recommended to control population quality but did not provide any definition of population quality.

2.     Improve the technology and foster green economy but did not mention the Chinese must successful “Grain for Green” program.

3.     Minimize the number of fossil fuel-based industries to regulate the carbon emission but did not provide any doable options.

4.     Regulate land use policies to sequester carbon but did not mention how to dovetail land use policies with the 21st century technology to meet the need for increasing population while minimizing the amount of carbon emission. Also, did not mention if there will be any impact of China’s recent reactions of not engaging in climate regulations [in protest of Nancy Pelosi’s visit to Tibet]

5.     Increased coordination among different government and non-governmental bodies to regulate the carbon emission.

The paper is well written, and deserves publications, however, there are some concerns. They need to be addressed thoroughly.

a.      Some empirical values ranges such as [0, 8] and references are put within the [ ] parenthesis, making it hard to differentiate between the references and empirical value ranges. Using [ ] to numerical values is confusing with references.

There are some concerns with the writing styles that need to be addressed.

Page 2, line 59: Sentence starting with “The basin….” How about stating the Yellow-River basin…”

Page 2, lines 64-65.

Page 2, line 87: “So far no unified method has been developed for the study of carbon emission [21]—really, is this statement correct?

Page 3: Line 116 (words jumbled together).

Page 3: Additionally, the majority of current research on the causes of land use carbon emissions concentrates on environmental and socioeconomic variables, frequently ignoring the influence of policy, technology, thought, and other human elements. Therefore, a more extensive study of land use carbon emissions needs to be performed, to improve public understanding of its causes.

I am curious why the authors have not dovetailed the issue of technology with human elements and Grain for Green Program to sequester carbon.

Page 3: Line 136—“Pacific River System” never heard off. Needs explanations what do the authors mean here.

Instead of using the term respectfully and put the land use types two sentences above the numerical values, it might make reading easier if the values of carbon emissions are tied to specific land use practices in sentences such as below.

arable land      182 0.0497 kg/(m2 ·a),

forest land       −0.0581 kg/(m2 ·a),

grassland         −0.0021 kg/(m2 ·a),

water area       −0.0253 kg/(m2 ·a),

unused land     −0.0005 183 kg/(m2 ·a)

Page 5, line 208 (comma after space).

Page 18, lines 398-403 need rephrasing.

Page 11: line 406, [0, 8] looks like reference.

Page 11, lines 409-413: How River basin pays attention, …..confusing sentence.

Page 11, lines 412-417: needs rephrasing.

Page 12, line 431—incomplete sentence.

Page 12, lines 431-434, better to rephrase.

Page 12, line 447…How, space needed after a period.

Page 13: line 501, reads awkward.

Page 13, line 505 redundant words.

Page 13: lines 508-509 need rephrasing for clarity.

Page 13 lines 515-516, need rephrasing for clarity.

Author Response

Dear reviewer,

Reviewer 2 Report

The authors have selected a very good area and topic for the research. Carbon emission from land use and land cover change is an important aspect towards global warming, which is an accepted phenomenon. I have studied the paper many times and tried to find out good reasons to accept the methodological aspect used in this paper. Apparently, the authors have tried to cover the Yellow River Basin in detail. Considering the technical aspects for the carbon accounting as being practiced in land use emissions / carbon sequestration vis-à-vis methodology used by the authors, I was thinking to reject the manuscript. However, I have decided to give the authors a chance to justify their efforts and if their reply sounds good then we may have detailed review round regarding results and discussion parts, though a few comments also jotted down to improve these parts as well. Following are key concerns which need to be settled first prior to move further for detailed round of review:

1.     The methodology section does not provide information on set of variables actually used for the estimation of emissions from land use practices vis-à-vis direct and indirect emission factors and scenarios as ambitiously discussed in first paragraph under section 4.1. Such a missing thing is also contradictory with the ambitious statement made by the authors in introduction section (lines 129-133) which needs to be revisited. Following actions are specifically required:

a.  There is a need to develop emissions scenario-based variables. For the purpose, first and foremost important thing to create in-depth understanding about all direct and indirect variables and then describe the methodology with clear steps followed.

b.  Develop and incorporate a methodological design and flow diagram in methodology section to illustrate the overall methodology used in the paper.

c.  The authors have confused heading for section 4. Revise the section heading. It can either be ‘Methodology’ or ‘Material and Methods’. Bring down section 3 i.e. study area by merging it with methodology section. So, the section number will also be reduced after merging the two.

2.     Equation 1 under section 4.1. is not described well. It is confused just with IPCC reference. There is a need to develop proper set of model equation after determining / describing detailed set of indicators against which quantitative data can be run in a statistical package. There is also needed to provide information about the statistical package with which this equation can be run.

3.     The authors have included Table 1 for the coefficients of carbon emissions. However, there is no reference and description on analytical review included in this regard, which need to be incorporated accordingly.

4.     At present Table 1 is standalone in the document. It is not clear how these coefficients were applied and on which dataset applied? It needs to be clearly mentioned in the methodology part and then linked up with the results section as well.

5.     Similarly, all equations in sections 4.3 and 4.4 are standalone and apparently there is no linkage with the variables and dataset which needs to be addressed properly.

6.     Section 4.5 mentions data sources. But, it is neither clear in this section nor any other part that what specific dataset (metadata or raw data) was acquired and used in the study? How are emission values calculated and reported in results section Table 2?

7.     Table 2: Values in Table 2 are different as mentioned in narrative form under section 5.1, which used a multiplier of 104. Secondly, it is not clear how these values are calculated or acquired. If calculated by the authors as part of the study, then methodological issue is there as discussed above, which needs to be clarified in detail. Besides, in case if these values are acquired then these must be cited properly, and methodology should be revisited accordingly. However, from the existing writeup, it seems difficult to decide on the situation. All other statistical tests are also linked with this clarity.

8.     There is no methodological limitation / shortcoming described in the paper. It needs to be added as sub-section of the methodology.

9.     There is a confusion about the Results and Discussion under one heading. Disintegrate both and make two separate sections i.e. one for Results and other for Discussion. Incorporate changes accordingly.

10.  What is knowledge addition with the outcome of this study? It needs to be incorporated in conclusion, after being discussed and justified with comparative referencing in discussion part.

Author Response

Dear reviewer,

Round 2

Reviewer 2 Report

Dear Authors, please see my review comments for round 2 in blue text:

Response to the reviewer 2

Thanks for your comments on our paper. We have revised our paper according to your comments:

Q1: The methodology section does not provide information on set of variables actually used for the estimation of emissions from land use practices vis-à-vis direct and indirect emission factors and scenarios as ambitiously discussed in first paragraph under section 4.1. Such a missing thing is also contradictory with the ambitious statement made by the authors in introduction section (lines 129-133) which needs to be revisited. Following actions are specifically required:

a.There is a need to develop emissions scenario-based variables. For the purpose, first and foremost important thing to create in-depth understanding about all direct and indirect variables and then describe the methodology with clear steps followed.

b.Develop and incorporate a methodological design and flow diagram in methodology section to illustrate the overall methodology used in the paper.

c.The authors have confused heading for section 4. Revise the section heading. It can either be ‘Methodology’ or ‘Material and Methods’. Bring down section 3 i.e. study area by merging it with methodology section. So, the section number will also be reduced after merging the two.

The authors’ answer: Firstly, we have divided Formula 1 into two formulas then describe the methodology with clear steps followed. Secondly, thank you for your suggestion to include the flow diagram. We have made significant changes to the methods section and explained in details and added the flow diagram. Finally, we have merged the study area with methodology section and revised the section heading to ‘Study Area and Methodology’.

Q1 - Review comment for round 2: Although effort has been done, para-wise comments not replied. Following are still needed:

a.      Heading for Section 3 should be ‘Material and Methods’

b.      Heading for section 3.1 is Okay.

c.      Heading for section 3.2 and its subsequent sub-headings are Okay.

d.      Added Figure 2 is not clear. Section numbers should not be part of the steps as shown in boxes for process flow followed. There is a need to improve the figure by bringing more clarity.

e.      A paragraph added vide line numbers 249 to 259 is standalone. There is a need to put a sub-heading for it i.e., 3.2.5 ‘Variables for the Study’.

Q2: Equation 1 under section 4.1. is not described well. It is confused just with IPCC reference. There is a need to develop proper set of model equation after determining / describing detailed set of indicators against which quantitative data can be run in a statistical package. There is also needed to provide information about the statistical package with which this equation can be run.

The authors’ answer: According to your suggestions, equation 1 under section 4.1. is not clear. We have divided Formula 1 into two formulas, with each formula explaining direct carbon emissions and indirect carbon emissions separately. Then we describe each word's significance in the formula and provide information about the statistical package with which this equation can be run.

Q2 - Review comment for round 2: Although effort is made, these need to be coherent with the variables added vide lines 249 to 259.

Q3: The authors have included Table 1 for the coefficients of carbon emissions. However, there is no reference and description on analytical review included in this regard, which need to be incorporated accordingly.

The authors’ answer: According to your suggestions, we have listed the standard coal conversion coefficients and carbon emission coefficients of different energy categories in Table 1. Below the new formula 2, we have added the reference and description on analytical review in this regard.

Q3 - Review comment for round 2: Although a reference is added in the text, there is need to bring clarity vis-à-vis variables concerned and added vide lines 249 to 259. It should be dealt along with my review comment for Q4.

Q4: At present Table 1 is standalone in the document. It is not clear how these coefficients were applied and on which dataset applied? It needs to be clearly mentioned in the methodology part and then linked up with the results section as well.

The authors’ answer: According to your suggestions, we have explained the coefficients and added the dataset in the methodology part. Details can be found in Equation 2 and below.

Q4 - Review comment for round 2: Not attended well. In lines 208 to 210, it is mentioned QUOTE“This study analyzes eight main energy sources: coal, coke, crude oil, diesel oil, kerosene, gasoline, electric power, and natural gas.” UNQUOTE, which is causing confusion vis-à-vis scope for the study and the variables described by the authors vide line numbers 249 to 259 as part of review round 1. These variables are contradictory at two places. Which one set of variables needs to be used. If the coal, coke, crude oil, kerosene, gasoline etc to be applied in this study then how they become relevant or develop interconnection with the variables described in lines 249 to 259? What is relevance of coal, coke, crude oil, kerosene, gasoline etc in the case of landuse emissions undertaken under the scope of this study? It is raising major concern on the overall methodology of the study. Given the level of confusion, it is not wise to allow publication of this paper in this form. Hence, the authors are advised to revise the entire manuscript and submit it again. I didn’t want but pushed to recommend decline of the paper at this stage. Authors may use this feedback for the improvement of the paper.

Q5: Similarly, all equations in sections 4.3 and 4.4 are standalone and apparently there is no linkage with the variables and dataset which needs to be addressed properly.

The authors’ answer: According to your suggestions, we have explained the variables and added the dataset in the original 4.3 and 4.4 sections. It is straightforward to understand and comprehend.

Q5 - Review comment for round 2: Not satisfactory.

Q6: Section 4.5 mentions data sources. But, it is neither clear in this section nor any other part that what specific dataset (metadata or raw data) was acquired and used in the study? How are emission values calculated and reported in results section Table 2?

The authors’ answer: The data sources are not clear in Section 4.5. According to your suggestions, we have moved the data sources from Section 4.5 to the corresponding part. For example, we have explained the independent variables and dependent variables in the GTWR model. In addition, the results in Table 2 were calculated by direct carbon emissions plus indirect carbon emissions.

Q6 - Review comment for round 2: Not satisfactory.

Q7: Table 2: Values in Table 2 are different as mentioned in narrative form under section 5.1, which used a multiplier of 104. Secondly, it is not clear how these values are calculated or acquired. If calculated by the authors as part of the study, then methodological issue is there as discussed above, which needs to be clarified in detail. Besides, in case if these values are acquired then these must be cited properly, and methodology should be revisited accordingly. However, from the existing write up, it seems difficult to decide on the situation. All other statistical tests are also linked with this clarity.

The authors’ answer: Firstly, values in Table 2 are different as mentioned in narrative form under section 5.1. We have changed the title of Table 2 by adding 104. Secondly, we have thoroughly revised the methodology section., as mentioned above.

Q7 - Review comment for round 2: Not satisfactory.

Q8: There is no methodological limitation / shortcoming described in the paper. It needs to be added as sub-section of the methodology.

The authors’ answer: According to your suggestions, we have added a discussion section at the end of the paper. The shortcomings of the method are described and analyzed in detail in the first paragraph of the discussion section.

Q8 - Review comment for round 2: Not satisfactory. How you can add discussion section after conclusion. It can be relocated by placing before conclusion and giving it heading as limitation of the study.

Q9: There is a confusion about the Results and Discussion under one heading. Disintegrate both and make two separate sections i.e. one for Results and other for Discussion. Incorporate changes accordingly.

The authors’ answer: According to your suggestions, we have disintegrated both and made two separate sections. Besides, we have added the analysis of the shortcomings of the method and the lack of research in the discussion section.

Q9 - Review comment for round 2: Not satisfactory. See line 260. It is still Results and Discussion. A wrong discussion section is created after the conclusion which is basically for the limitation of the study and needs to be relocated. Whereas my original query needs to be addressed carfefully.

Q10: What is knowledge addition with the outcome of this study? It needs to be incorporated in conclusion, after being discussed and justified with comparative referencing in discussion part.

The authors’ answer: According to your suggestions, we have added the discussion in the end of paper. The discussion section focuses on the limitations and knowledge addition that justified with comparative referencing

Q10- Review comment for round 2: Not satisfactory.